# Vulnerability Analysis Method Based on Network and Copula Entropy

**DOI:** 10.3390/e26030192

**Published:** 2024-02-23

**Authors:** Mengyuan Chen, Jilan Liu, Ning Zhang, Yichao Zheng

**Affiliations:** 1School of Finance, Central University of Finance and Economics, Beijing 102206, China; 18902755229@163.com (M.C.); liu_jilan@163.com (J.L.); 2022210315@email.cufe.edu.cn (Y.Z.); 2China Fintech Research Center, Central University of Finance and Economics, Beijing 102206, China

**Keywords:** graph theory and network analysis, copula entropy, market vulnerability

## Abstract

With the deepening of the diversification and openness of financial systems, financial vulnerability, as an endogenous attribute of financial systems, becomes an important measurement of financial security. Based on a network analysis, we introduce a network curvature indicator improved by Copula entropy as an innovative metric of financial vulnerability. Compared with the previous network curvature analysis method, the CE-based curvature proposed in this paper can measure market vulnerability and systematic risk with significant advantages.

## 1. Introduction

As the “blood supply station” of the modern economy, finance has become increasingly vulnerable in the face of rising global inflationary pressure and the frequent occurrence of related crises, which gradually endanger the security of the real economy. Many countries are increasingly concerned about how to prevent financial risks and measure market vulnerability. Along with the continuous progress of technological tools, new methods of measuring financial market vulnerability are urgently needed to enable countries to respond to and prevent crises in a timely manner and ensure smooth economic operations.

In a narrow sense, financial vulnerability is considered to be the inherent instability caused by the high debt operation of the financial industry itself [1]. Huang put forward a broad definition of financial vulnerability considering external shocks on this basis, thinking that it generally refers to the accumulation of risks in all financing fields, including credit financing and financial market financing [2]. At the same time, the author also pointed out that the financial market’s vulnerability comes from the volatility of asset prices and the linkage effect of volatility. Based on this general definition, different scholars have tried to study financial vulnerability from multiple perspectives, such as the banking system, stock market, and futures market, and think about how to use management methods and measurement methods to deal with risks, such as portfolio investment [3,4,5].

Based on the goal of preventing risk accumulation and crisis occurrences, both domestic and international scholars have begun researching how to measure financial market vulnerability. However, due to the numerous factors that affect financial vulnerability and the lack of a unified definition, the corresponding measurement methods are also diverse. Currently, common measurement methods include (1) constructing financial crisis early warning models, such as the signal model created by Kaminsky et al., which assesses the crisis based on the comparison between different financial variables and their critical values at a given moment [6], and (2) constructing measurement indicators, such as the one created by Lin, who used an APARCH model to characterize the asymmetric volatility of the conditional volatility of financial returns [7]. Wang attempted to measure the degree of financial vulnerability from four dimensions: the macroeconomy, banking system, stock market, and real estate market [8]. Spelta et al. considered the endogenous instability caused by the “herding effect” and positive feedback and studied financial market vulnerability through quantifying the intensity of self-organizing processes arising from stock returns’ comovements and self-similarities [9].

Unfortunately, despite the increasing sophistication of models and indicator methods, there are still problems, like the inability to dynamically demonstrate changes in market vulnerability and characterize market risk contagion. With the rapid development of complex network analysis methods and tools, these methods are increasingly used in financial research, such as equity structure research [10,11], risk contagion research [12,13], and so on. And, newer studies on financial vulnerability measures have also started to introduce weighted undirected graphs as abstract models of the financial system [14,15,16,17,18,19]. Based on this theory, Sandhu et al. proposed using the curvature index to measure financial vulnerability [20], but the related research is relatively preliminary and the research field of applying various discrete curvatures to measure market vulnerability still needs to be further studied.

An important drawback of previous network analysis methods is that they often use linear correlation to describe the dependencies and associations between nodes while ignoring higher-order correlations, and these approaches are often tepid in measuring tail risk. A good example is the 2008 financial crisis, when the Gaussian Copula function, which measures linear correlation, was widely used in derivative pricing. However, linear correlation ignores the higher-order correlations between variables, especially tail risks. Therefore, when a series of default events broke out collectively, it was finally found that derivatives constructed by linear correlation lost the function of risk hedging. Compared with using traditional methods such as the Pearson or Spearman correlation coefficient to measure the correlation between nodes in a network, we attempt to apply the Copula entropy (CE) proposed by Ma and Sun to network construction [21]. CE can measure the statistical correlation of all orders, which can provide a clearer correlation pattern for the network and show a strong correlation measurement effect, so it is suitable for this study. At present, some scholars have used the measurement advantages of CE to conduct network research in the field of hydrology, but economics-related research is still relatively scarce, and this paper can provide some supplements [22,23,24].

The following is the arrangement of each part of this paper: Section 2 and Section 3 outline our method; Section 4 is about our empirical results and analyses; and Section 5 outlines our conclusions.

## 2. Copula Entropy and Ricci Curvature

### 2.1. Nonlinear Causal and Information Captured with Copula Entropy

Correlation analysis is a pivotal concern in multivariate financial analysis, encompassing aspects such as asset pricing, investment portfolios, the transmission and spillover of volatility, and risk management. However, conventional linear correlation coefficients possess certain limitations as they are more suitable for Gaussian distribution data and typically necessitate the linearity of variables and the existence of variance. In practical terms, numerous data observed in financial markets often exhibit peaked distributions with heavy tails, while the presence of heteroscedasticity frequently renders variance nonexistent. Traditional financial modeling tools fail to adequately accommodate these modeling requirements.

The Copula theory proposed by Sklar in 1959 [25] solves the modeling problem in traditional financial analysis while also effectively characterizing the nonlinearity, asymmetry, and tail correlation relationships between variables and financial data. Usually, the marginal distribution of each variable can be easily determined from the joint distribution of random variables, but it was difficult to determine the joint distribution from the marginal distributions. The Copula theory shows that any correlation between multiple variables corresponds to a function used to represent this relationship, called the Copula function, which connects the joint distribution of multiple random variables with their respective marginal distributions. It can decompose a multivariate joint distribution into multiple marginal distributions and a Copula product and capture the correlation between multivariate variables through joint functions. With the continuous development of the theory in recent years, it has become an important tool for asset correlation analysis. Li and Shi [26] empirically showed that the tails of financial assets usually exhibit asymmetric correlations, and the Copula function better reflects the correlation structure of such assets. Jondeau and Rockinger [27], Wen and Feng [28], and Virbickaitė et al. [29] also fully proved this point.

Given the joint distribution function *F(**X**)*, the marginal distribution function $F_{i}(X_{i}$*)*, and the Copula function *C(**u**)* of any *N*-dimensional random variable ***X***, the joint distribution function can be expressed as the form of the Copula function with the inputs being the marginal distribution functions, as Equation (1):(1)$$\begin{array}{c} F\left(X\right)=C\left(F_{1}\left(x_{1}\right),\cdots ,F_{N}\left(x_{N}\right)\right) \end{array}$$

Since mutual information has always been considered to measure all correlation information, Ma and Sun proved the correlation between the Copula function and mutual information in 2008, and thus proposed the concept of Copula entropy [21].

As we know, previous network analysis methods based on traditional correlation coefficients are only able to describe linear correlations between nodes, which is not sufficient for our study of financial markets with complex tail correlations. Therefore, we introduce Copula entropy to improve the method.

Copula entropy was proved to be equivalent to mutual information in information theory [30]. Essentially, it is a form of Shannon entropy, expressed as
(2)$$H_{c}\left(u_{1},\cdots ,u_{n}\right)=-\int_{0}^{1}\cdots \int_{0}^{1}c\left(u_{1},\cdots ,u_{n}\right)\log\left[c\left(u_{1},\cdots ,u_{n}\right)\right]du_{1}\cdots du_{n}$$
Considering the case of two variables, Equation (3) is expressed as
(3)$$-H_{c}\left(u_{1},u_{2}\right)=H\left(X_{1}\right)+H\left(X_{2}\right)-H\left(X_{1},X_{2}\right)$$
Among them, $H\left(X_{i}\right)\left(i=1,\;2\right)$ and $H\left(X_{1},\;X_{2}\right)$ are the uncertainty degree of the random variable $X_{i}$ obtained by Shannon information theory and the joint entropy of the random variable $X_{1}$ and $X_{2}$, respectively, which are calculated by Equation (4) and Equation (5), respectively:(4)$$H\left(X\right)=-\overset{N}{\underset{i=1}{\sum }}p\left(x_{i}\right)logp\left(x_{i}\right)$$
(5)$$H\left(X,Y\right)=-\overset{N}{\underset{i=1}{\sum }}\overset{N}{\underset{j=1}{\sum }}p\left(x_{i},y_{i}\right)log\;p\left(x_{i},y_{i}\right)$$
In addition, mutual information can be regarded as the amount of information contained in a random variable about another random variable or the uncertainty of a random variable that is reduced by knowing another random variable, expressed as Equation (6):(6)$$\begin{array}{c} M\left(X,Y\right)=\overset{N}{\underset{i=1}{\sum }}\overset{N}{\underset{j=1}{\sum }}p\left(x_{i},y_{i}\right)log\frac{p\left(x_{i},y_{i}\right)}{p\left(x_{i}\right)p\left(y_{i}\right)}=H\left(X\right)+H\left(Y\right)-H\left(X,Y\right) \end{array}$$
Obviously, $H_{c}\left(u_{1},u_{2}\right)=-M\left(X_{1},X_{2}\right)$, that is, Copula entropy, has the same value as interaction entropy with an opposite sign [30,31]. Copula entropy is mutual information.

In the context of correlation becoming an important indicator of financial market risk, the advantages of CE over the traditional Pearson coefficient are as follows: (1) The Copula function value can be obtained by using a nonparametric estimation method, which does not require random variables to conform to or be close to normal distribution, and works well when measuring the correlation of nonelliptic distribution families. (2) Different Copula functions can be selected to portray the correlation between variables according to the focus of the study, including the Gumbel Copula function, which is sensitive to the change in the upper tail of the variable distribution; the Clayton Copula function, which is sensitive to the change in the lower tail; and the Frank Copula function, which is not sensitive to the change in the tail, etc. (3) It is not limited by dimensions and can measure the correlation between multidimensional variables, including nonlinear relationships.

In fact, Copula entropy is not the first optimization of the Pearson coefficient. In 1904, the British psychologist Spearman proposed the rank correlation coefficient [32]. In the statistical sense, this coefficient can be regarded as a special case of the Pearson correlation coefficient, but the main difference is that it does not require variables to be close to or conform to a normal distribution, and it can measure the nonlinear relationship between two variables. However, the Spearman coefficient also has the disadvantage that it cannot reflect the correlation structure of variables and cannot calculate the correlation of multidimensional variables. Therefore, this paper finally chooses Copula entropy to measure the correlation and correlation structure between variables and compares the empirical effect of traditional correlation coefficients and CE on the CSI 300 data in Section 4.1 to verify the above conclusions.

### 2.2. Market Vulnerability Measurement with Ricci Curvature

Although graphs and networks are composed of discrete objects, they can be considered as metric spaces just like smooth manifolds, where the distance between any two nodes can be specified by the length of the path between them. The classic Ricci curvature is applicable to smooth manifolds and requires tensor and higher-order derivatives, and thus it cannot be directly applied in discrete graphs or networks. In order to make Ricci curvature applicable to financial vulnerability studies, it is necessary to focus on the quantification of Ricci curvature for two basic geometric properties of manifolds, namely the bulk growth and divergence of geodesics. In n-dimensional Riemannian manifolds, Ricci curvature controls the n−1 dimensional volume growth of geodesics along a certain vector direction in n-dimensional stereo angles; in addition, Ricci curvature quantifies the dispersion of geodesics with the same initial point in a given stereo angle.

Classic Ricci curvature is related to vectors in smooth manifolds, and extended to network analysis, the notion of discrete Ricci curvature is related to edges rather than vertices or nodes. Since the discretized Ricci curvature does not have all the characteristics of the classic Ricci curvature, different discretization methods can give different interpretations to the graph or network and then produce different curvature concepts, such as Ollivier–Ricci (OR), Forman–Ricci (FR), Menger–Ricci (MR), Haantjes–Ricci (HR), etc. The OR curvature realizes discretization processing based on classic Ricci curvature by comparing the optimal average distance and Euclidean distance between neighbor nodes, and its core process lies in the solution of the optimal average distance; FR curvature is mainly based on the relationship between Laplacian and Ricci curvature, which is more algebraic in nature and can quantify the amount of information propagated by the end of the edge in the network; the basic principles of HR curvature and MR curvature are relatively similar, both derived from the definition of measuring triangle curvature proposed by Menger. The main difference is that MR curvature only considers triangles or simple paths of length two formed between two nodes of an edge, while HR curvature considers longer paths between two nodes.

There are different curvature properties in the above four discrete Ricci curvatures due to differences in their principles and metrics. In a discrete network, the Ollivier–Ricci curvature can well explain the volume growth in the classic Ricci curvature, while the Forman–Ricci, Menger–Ricci, and Haantjes–Ricci curvatures describe the divergence of geodesics in the classic Ricci curvature characteristic. The specific introduction of the four curvatures is available in Appendix A.

## 3. The Calculation of Curvature

In general, our analysis methods are as follows: (1) We calculate the CE between assets in the market by referring to the method of Ma and Sun [21]. In this step, we try to account for the superiority of CE compared to the linear correlation coefficients. (2) We add CE into the construction of the market network to obtain the curvature measure we need.

In the first step, we use the nonparametric CE estimation method [33]. The method consists of two steps: (1) estimate the empirical Copula density function and (2) estimate the CE with the empirical Copula density function.

Given a set of independent identically distributed samples $\left\{x_{1},\ldots ,\;x_{T}\right\}$ of random variable *X*, the empirical Copula density function can be easily estimated by ordinal statistics, as Equation (7):(7)$$F_{i}\left(x_{i}\right)=\frac{1}{T}\overset{T}{\underset{t=1}{\sum }}1\left(x_{t}^{i}<x_{i}\right)$$
where 1(·) represents the indicative function.

After obtaining the empirical Copula density function, we used the K-nearest neighbor method proposed by Kraskov et al. to estimate the CE [34].

In the second step, we do not directly use the CE as the edge weight, but we treat it as follows: We use the following processes on the calculated CE according to its distribution: Tail the CE of the perfectly positive correlated stocks to two; then, use the linear scaling method to transform the scale of the CE to [0, 1]. The purpose of these processes is to enlarge the difference between the CE in the value range, thus constructing a heterogeneous network as much as possible. On this basis, Equation (8) is used to measure the distance between individual stocks as the weight of the edge:(8)$$w=2\times \sqrt{1-adj\_CE}$$

In order to extract the main information in the network, a threshold α=0.5 is set according to the distribution of the CE. All edges with adj_CE below the threshold will be deleted from the network. The setting of the threshold takes into account not only the distribution of the CE but also the computational efficiency and effectiveness.

## 4. Empirical Results and Analyses

### 4.1. CE and Correlation Coefficient

In order to explain the role of CE in correlation measurements more deeply, in this part, we explore the characteristics of CE based on the data of constituent stocks within the CSI 300 index. First, we try to compare CE with two common correlation coefficients, the Pearson correlation coefficient and the Spearman correlation coefficient. The former is usually used to measure linear correlation, while the latter, which is a nonparametric version of the former, can measure partial nonlinear relationships.

Therefore, we calculate the correlation coefficients and CE between every two component stocks and output the matrix as shown in Figure 1. In order to show the correlation between stocks more clearly in the figure, we group the stocks of the same industry together according to Shenwan’s industry classification standard. As can be seen from the figure, CE provides a clearer correlation pattern than these two correlation coefficients. This clarity is reflected in the fact that the heat map based on CE exhibits deeper colors in regions where we expect a strong correlation, while extremely light colors appear in regions where we expect no strong correlation. We will find that the areas with deep color in the graph are usually very concentrated, which is precisely because we grouped stocks of the same industry together when arranging the coordinates. Unlike CE, the heat maps based on correlation coefficients show deep colors that are irregular even in regions where a strong correlation is not expected. That is, in the regions with a low correlation, the two correlation coefficient measures will have more noise, resulting in the correlation of the truly correlated region not being clear in the matrix. This is because CE measures statistical correlations for all orders, whereas correlation coefficients only measure statistical correlations for second orders. In the internal comparison of the two correlation coefficients, the color difference in the Spearman’s correlation coefficient matrix is slightly more pronounced than that of Pearson’s, probably because it is also estimated nonparametrically. However, there is still a big difference between the former and CE. Theoretically, they use different Copula functions. CE can describe a statistical correlation more clearly than the Spearman correlation coefficient.

In Figure 1, we use the blue-border-boxed area to illustrate the superiority of CE. In the example area, the heatmap based on the correlation coefficient shows a deep color, suggesting a certain positive correlation. However, in the context of CE, the heatmap shows a light color, indicating a weak correlation. The specific correlation strength is as shown in Table 1’s Panel A, B, and C, where the correlation coefficients are around 0.4 and the CE values are all zero (negative because of the calculation error). From an economic perspective, the two groups of stocks with correlation noise are the metallics and basic chemical industries, which should not have too much correlation in the market. Therefore, the CE measure is more reasonable.

In addition, there is an obvious difference between CE and the normal correlation coefficient; that is, CE measures the correlation strength between variables and does not distinguish between positive and negative, while the correlation coefficient measures the positive and negative correlation between variables and takes a value within [−1, 1].

Despite CE’s inability to distinguish the direction of the correlation, its application to distance measurements in security market networks is reasonable because the security market does not have a significant negative correlation. This means that the strong correlations we identify through CE are positive in the vast majority of cases. Moreover, in our treatment of the edge weights, we remove those with a smaller CE, such that the final edges in the network represent a stronger positive correlation. Therefore, the weights of the edges in the final network actually represent the strength of the positive correlation.

### 4.2. Network Analysis with CSI 300 Component Stocks

On the basis of exploring the Copula entropy of CSI 300 component stocks in the previous section, we attempt to construct a market network composed of component stocks in CSI 300 from April 2006 to April 2022. We adopt a similar analysis route to Samal et al. [35], and our main contribution is the proposal of CE-based Ricci curvature. Therefore, we calculate the changes in the following four kinds of curvature over time in the network: Ollivier–Ricci curvature, Menger–Ricci curvature, Haantjes–Ricci curvature, and Forman–Ricci curvature. In addition, we compared the CE-based approach with the correlation-coefficient-based approach. The stock data used in this section and subsequent sections are from the CSMAR database. The time windows used in the calculations are 20 trading days.

Figure 2 reports the network curvature based on CE for the period from April 2006 to April 2022. From top to bottom is the Ollivier–Ricci curvature, Menger–Ricci curvature, Haantjes–Ricci curvature, and Forman–Ricci curvature. The two shaded sections mark the periods when stock market crashes occurred in 2008 and 2015. In addition, the curvature based on Pearson’s coefficient of the network is also reported, which is identified by a light orange curve in the figure.

From the perspective of measuring market vulnerability and systematic risk, the curvature measure undoubtedly fulfilled the task well: all four curvatures remained high for a long period of time after the crisis broke out (Forman curvature is always negative, and the greater the absolute value, the more fragile the network structure). Enduring a bear market is often painful, so the curvature measure gives a “fragile and dangerous” signal for a long time after the crisis. This phenomenon can be explained by the noise-trader model that De Long et al. put forward [36]. The model said that the transaction noise brought by extreme emotions is difficult to eliminate, so it often leads to long-term value deviation. In addition, this phenomenon may also be closely related to the arbitrage asymmetry proposed by Stambaugh et al. [37]. The wild fluctuations in the market lead to the increase in arbitrage risk, so the deviation in the value is more serious. In short, the mechanism of “vulnerability” in the security market is not exactly the same as the transmission of risk when a banking crisis occurs. It should be noted that the measurement method proposed in this paper itself cannot predict the arrival of the crisis in a forward-looking way. It is not difficult to see from the figure that the curvature is essentially a signal that exists simultaneously with the crisis, and it measures the state of the market. However, after the crisis occurs, this index can indicate when the market comes to normal, which is also quite important.

What is more, Figure 2 also shows the curvature sequence of the network constructed by Pearson’s correlation coefficient, which is calculated with the same method as [35,38]. We find that it has a similar trend to the CE-based method. However, the CE-based method provides more pronounced differences between different market environments, indicating its superior ability to capture market vulnerability and differentiate the magnitude of systematic risk. Visually, this feature is represented by higher curvature peaks during crisis periods and comparatively smaller curvature measurements during other stable periods. Additionally, during the crisis period (gray area in Figure 2), the CE-based index rises faster, which indicates that it can better reflect the sharp release in market risk. It is notable that, immediately after the crisis period, the Pearson-based curvature shows a brief but sharp dip and then goes back to a relatively high position. It is a typical fake stabilization. By contrast, the CE-based index is more robust in measuring risk by remaining at a higher level of curvature, which provides more cautious risk monitoring after the event.

In addition to marking the special time period of 2008 and 2015, we further explore the network structure characteristics at five key time nodes: 28 January 2008, 15 April 2011, 3 January 2014, 26 June 2015, and 17 February 2017. The community discovery algorithm is used to realize the modularization of the network structure, as shown in Figure 3. The five figures above the curvature diagram represent the graph network model formed by the component stocks of CSI 300 at different time points. After modularization, the color of the nodes represents the module (community) to which they belong.

The modularity measure Q, also known as the modularity degree, is a commonly used method to measure the structural strength of network communities. It was first proposed by Mark NewMan. The modularity degree Q mainly depends on the community division in the network, so we use Q to quantitatively measure the division quality of network communities. As Figure 3 shows, during the two stock market crash periods, with four kinds of curvature at a high level, there are significantly more edges in the network. As a whole, the modularity levels of networks are relatively low (the measure of the modularity Q is between 0.1 and 0.2), and we could not distinguish different communities clearly. In contrast, during the 13 January 2014 and 17 February 2017 periods with low curvatures, the number of edges formed in the network is significantly less than that in other periods. At the same time, there is a higher modularity level (the modularity degree Q is higher than 0.7), and the characteristics of community distribution morphology and scale can be clearly observed from the network. In addition, although 15 April 2011 is not the peak of the curvature in the interval, the curvature fluctuates greatly. With the large number of edges and low modularity in the network, at this time, the market usually has high vulnerability and systematic risk. According to the above results and analyses, it can be preliminarily inferred that in general, high curvature is accompanied by a low module degree of the network structure, large number of edges, close connections between different nodes, and high market risk. Low curvature corresponds to a high module degree, fewer edges, less closeness between nodes, and less market risk and vulnerability.

### 4.3. Comparison with Traditional Risk Metrics

In order to reveal the characteristics of network curvature in measuring financial risk, we select several risk metrics commonly used in finance for comparison, which are the volatility of the optimal portfolio of risky assets, the volatility of the index estimated by the GARCH model, and the realized volatility of the index. It should be noted that the above three metrics mainly indicate risk from the market. Market risk is a type of systematic risk, and only in the framework of the CAPM does systematic risk include only the component from the market. But to a certain degree, these metrics are closely related to systemic risk.

In this paper, the optimal risky asset portfolio is computed by using Markowitz’s framework to maximize the expected utility of an investor with a quadratic utility function. In Markowitz’s framework, the optimal risky asset portfolio is the market portfolio with a *β* of 1, whose volatility measures the magnitude of market risk. In calculating the optimal risky asset portfolio, this paper uses the maximization of the Sharpe ratio, which is maximizing $\frac{r_{p}-r_{f}}{\sigma_{p}}$. In this formula, $r_{p}$ and $\sigma_{p}$ are the portfolio’s average daily return and volatility (standard deviation of returns) for the year, respectively, and $r_{f}$ represents the year’s risk-free rate of return.

Index volatility is a good indicator of market risk. As a commonly used regression model to analyze financial data, GARCH can more accurately simulate the volatility changes in time series variables than ARCH. More specifically, the GARCH(1, 1) model is the most widely used. This paper takes the daily return data of the CSI 300 index as the research sample and establishes the GARCH(1, 1) model to analyze the return volatility. By drawing the time series chart, as shown in Figure 4, it can be seen that the volatility of return in this period of time is large and there is the phenomenon of “volatility aggregation”. Additionally, the ADF test and correlation test show that the return series is smooth without significant correlation. On this basis, we use ARMA(1, 1) to fit the yield series. The residuals passed the ARCH effect test, which is a prerequisite for GARCH modeling.

The realized volatility, which is the historical volatility of an asset, is a commonly used risk measure. Like the index volatility estimated by the GARCH model, the realized volatility of an index can be used to measure the overall market risk. In this paper, we calculate the realized volatility of the CSI 300 index for each month (20 trading days).

Figure 5 illustrates the trend of the four curvature measures and the three commonly used risk metrics above. It shows that the seven risk metrics are extremely similar over time. In addition, Table 2 reports the results of the Pearson correlation coefficient tests for the seven risk metrics. From top to bottom, there are the Ollivier–Ricci, Menger–Ricci, Haantjes–Ricci, and Forman–Ricci curvatures as well as the volatility of the optimal risk portfolio, the volatility estimated by the GARCH(1, 1) model, and the realized volatility. There are highly significant correlations among the seven metrics, and this result suggests the commonality of the seven risk measures, as well as the robustness of the CE-based network analysis approach. Furthermore, in the CE-based network analysis approach, the amount of information is relatively large, which may identify tail risks that cannot be identified by the other three traditional indicators. Accordingly, the superiority could be seen in the period of 2008~2015.

Since the latter three risk metrics (the volatility of the optimal risk portfolio, forecasted volatility of the index, and realized volatility) are all representative of systematic risk from the market, it is feasible to use curvatures to measure the systematic risk empirically.

### 4.4. Ability to Explain Returns

The above analysis has shown some properties of network curvature based on CE. To more fully illustrate the superiority of the CE method compared to the correlation coefficients, in this section, we compared the ability of the two curvature measures to explain excess returns. According to modern asset pricing theory, if a factor has a stronger ability to explain excess returns, it is more efficient. Additionally, the ability to explain excess returns is the most intuitive financial meaning of this vulnerability metric. To this end, this paper explores the ability of curvature as a macro factor to explain stock excess returns, following the approach used by Fama and MacBeth (1973) [39]. Specifically, we add curvature to the CAPM model and Fama–French three-factor model to reform a two-factor model and a four-factor model, providing explanations about the variation in stock excess returns in the corresponding intervals. In this part, all the variables used are shown in Table 3, and all the data are from the CSMAR database.

In this paper, 1245 stocks listed in the A-share market before 1 January 2006 (2006-01-01) are selected to construct an investment portfolio by using the independent double-sorting method. The necessity of constructing the portfolio is to diversify the nonsystematic risk as much as possible, thus enhancing the explanatory power of the factors. To be specific, we use data from 1 January 2006 (2006-01-01) to 31 December 2010 (2010-12-31) to calculate the factor exposure of individual stocks on market return and curvature (Ollivier–Ricci in this part) separately, and then divide them into 20 groups according to the factor exposure in each factor dimension independently, resulting a total of 400 portfolios. Then, we apply Fama–MacBeth regression on the data from 1 January 2011 (2011-01-01) to 30 April 2022 (2022-04-30) to explore the explanatory power of the curvature on returns.

Fama–MacBeth regression is a cross-sectional approach, and its main process can be divided into two steps: firstly, a rolling regression on time series is conducted to calculate the factor exposure of the portfolio excess returns on each factor, and secondly, at each time point, a cross-sectional regression of the factor exposure, obtained in step 1, on the excess returns is conducted to observe their relations and examine whether there is a pricing error. The specific models are as follows (Equations (9) and (10)):(9)$$R_{i,t}=\alpha +\beta_{1,i,t}C_{t}+\beta_{2,i,t}RM_{t}+\varepsilon_{i,t}\;\;i=1,\;2,\;\ldots ,\;N$$
(10)$$\begin{array}{c} R_{i,t}=\alpha +\lambda_{1}\beta_{1,i,t}+\lambda_{2}\beta_{2,i,t}+\varepsilon_{i,t}\;\;t=1,\;2,\;\ldots ,\;T \end{array}$$

Regarding the time series regression, this paper takes a 60-month time window for rolling regression according to model (9); $C_{t}$ is the curvature and $RM_{t}$ is the market excess returns, which is proxied by the excess return of the CSI 300 index relative to the risk-free rate. It should be noted that, in order to be uniform with the curvature metric, 20 trading days are seen as 1 month here. The model of cross-sectional regression is shown in (10); $\beta_{1}$ is the factor exposure of the curvature and $\beta_{2}$ is the factor exposure of the market return.

Table 4 reports the results of the above two-factor Fama–MacBeth regression model. Columns 1–4 of the table report the explanatory power of the CE-based vulnerability measure for the returns while columns 5–8 report the results based on the Pearson coefficient. We explore the ability of the Ollivier–Ricci, Menger–Ricci, Haantjes–Ricci, and Forman–Ricci curvatures to explain market excess returns. It is found that when using CE, the relationship between factor exposure and the excess returns is significant no matter which measure of curvature is used; that is, it could explain the changes in the returns. Moreover, the hypothesis that the constant in regression is equal to zero cannot be denied, suggesting that there is no large pricing error, which strongly supports the financial significance of the curvature.

The CE-based method has stronger explanatory power for excess returns, which confirms our previous discussion on CE and correlation coefficients. Specifically, in the cross-sectional regression reported in columns 5–8, the significance of the Pearson coefficient for the curvature factor exposures is lower than those in columns 1–4. At the same time, the significance of the market factor exposures and constant term coefficients increased with the correlation-based method. This indicates that the CE-based method can obtain more effective vulnerability measures and better quantify market conditions.

For robustness, we also attempt to add the curvature measurement to the Fama–French three-factor model to further compare the explanatory power of the two vulnerability measurements for excess returns. Subsequently, we categorize stocks into four groups based on their factor exposures across each dimension, resulting in a 4×4×4×4 classification scheme. Finally, employing Models (9) and (10), we apply the Fama–MacBeth regression analysis on post-2011 data.

Table 5 reports the results of the above four-factor Fama–MacBeth regression model. Consistent with above, columns 1–4 of the table report the explanatory power of the CE-based vulnerability measure for returns while columns 5–8 report the results based on the Pearson coefficient. It is found that when using CE, the relationship between factor exposure and the excess returns is significant unless the Ollivier–Ricci curvature is used. That is, in the four-factor model, most curvature based on CE could explain the changes in returns. There is little pricing error, which strongly supports the robustness of the financial significance.

Additionally, the Pearson coefficient is employed to obtain results. As depicted in columns 5–8 of Table 5, there is a substantial reduction in the explanatory power of factor exposures toward excess returns across all four curvatures examined. Simultaneously, it is noteworthy that the t-value associated with the constant term exhibits a significant increase, which suggests an augmented likelihood of mispricing.

In the following part, we change the sorting method in the combination construction process into conditional sorting; that is, after the first dimension grouping is completed, within each group, the next dimension is sorted and grouped, and so on. We present the results of the Fama–MacBeth regression in Table 6. The two-factor model is used in rows 1 and 2 and the four-factor model is used in rows 3 and 4. The first row is grouped according to the curvature first and then according to the market factor; the second row is grouped according to the market factor first and then according to the curvature; the third row is grouped according to the curvature, market factor, SMB, and HML at once; and the fourth row exchanges the order of the curvature and the market factor. The results reported in Table 6 support the ability of the vulnerability measurement to explain excess returns and the superiority of the CE-based method over the Pearson coefficient.

## 5. Conclusions

This paper introduces Copula entropy into a curvature analysis of the network and generates some insights into measuring the vulnerability and systematic risk of financial markets. We find that CE provides a clearer identification of correlations than the Pearson and Spearman correlation coefficients, presenting less noise in the low correlation region. CE shows a better correlation measurement performance and can effectively distinguish different financial sectors. This is due to the fact that CE measures statistical correlations of all orders while the correlation coefficient measures only the second-order statistical correlations.

Based on the network analysis, the curvature trends are similar for both correlation measurement methods, but the CE approach is relatively more distinct across market states and has a greater ability to characterize market vulnerability and distinguish the magnitude of systematic risk in the market. Furthermore, the CE-based approach has been proven to have a stronger ability to explain excess returns, which verifies the superiority of the CE-based vulnerability measurement from another perspective. The figures show that while the curvature metric cannot predict the onset of a crisis in advance, it remains high for a long time after the outbreak of a crisis, giving a signal of vulnerability and danger. Therefore, it can suggest when the market will start to stabilize. In addition, high curvature is often associated with low modularity and a high number of edges in the network structure, with close ties among different nodes and high market risk. Accordingly, low curvature corresponds to high modularity and a low number of edges in the network structure, with reduced closeness among nodes, moderated market risk, and low vulnerability.

## Figures and Tables

**Figure 1 entropy-26-00192-f001:**
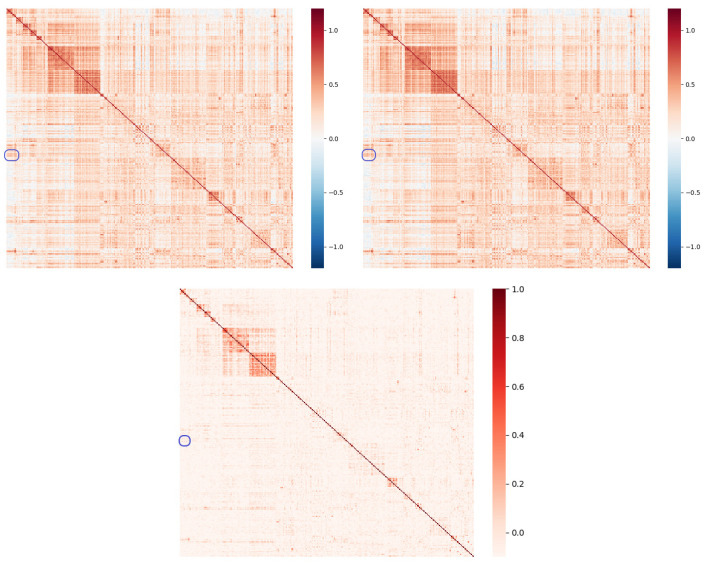
Pearson (**upper left**) and Spearman (**upper right**) correlation coefficient and CE (**bottom**). The coordinate axis of the figure represents the components of the CSI 300 index, and we rearranged these stocks according to the industry (from Shenwan) on the coordinate axis.

**Figure 2 entropy-26-00192-f002:**
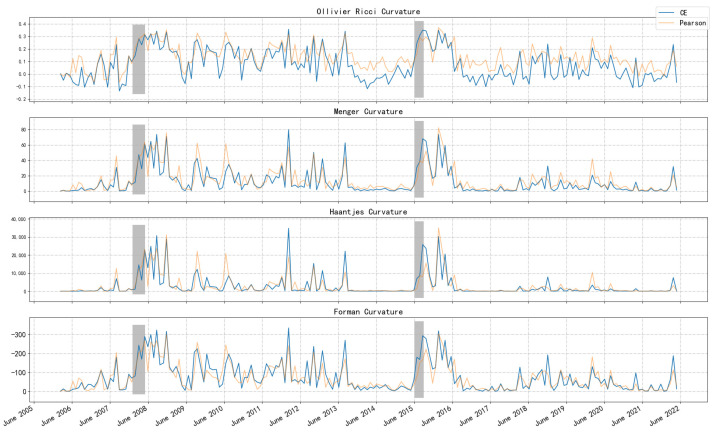
Curvatures of CSI 300 network.

**Figure 3 entropy-26-00192-f003:**
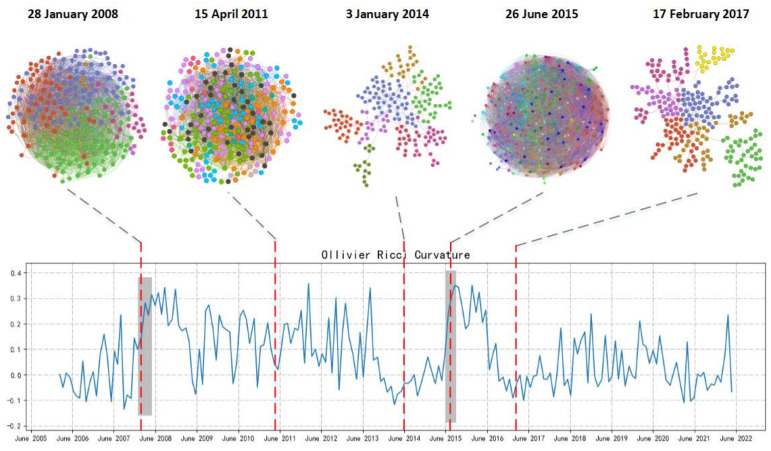
Curvature diagram and modularization of network.

**Figure 4 entropy-26-00192-f004:**
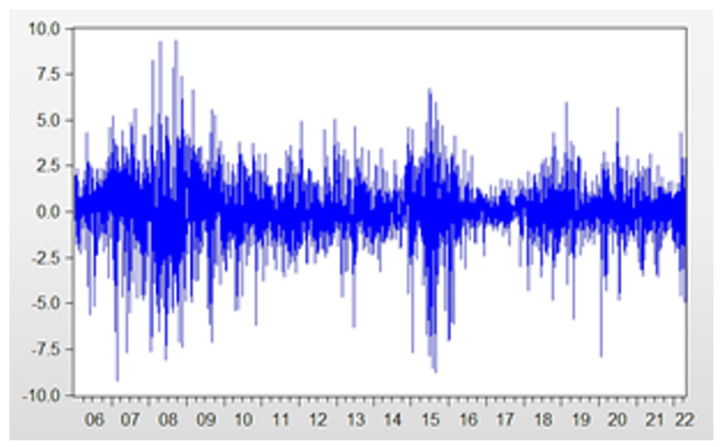
Time series of CSI 300 index returns.

**Figure 5 entropy-26-00192-f005:**
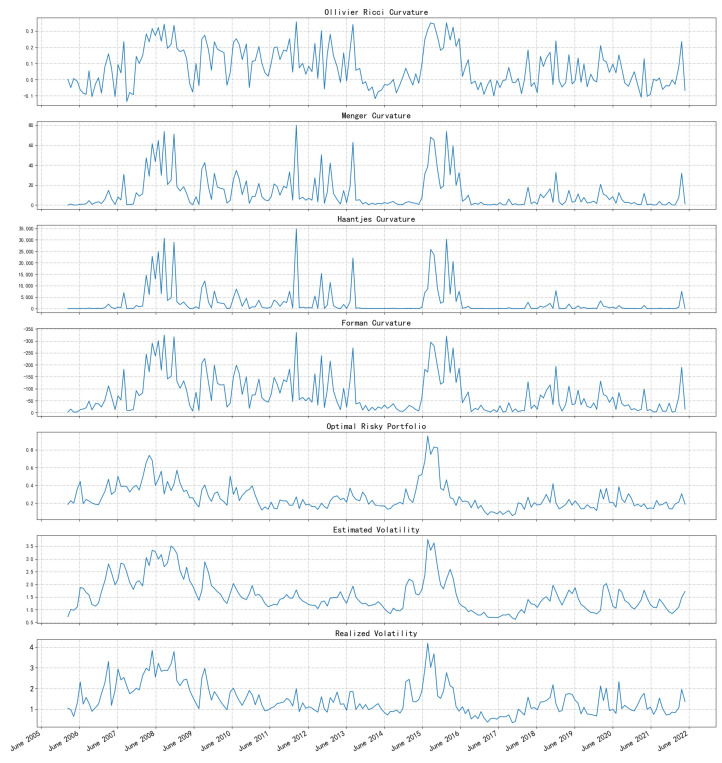
Seven financial risk metrics. From top to bottom, the Ollivier–Ricci, Menger–Ricci, Haantjes–Ricci, and Forman–Ricci curvatures are shown, as well as the volatility of the optimal risk portfolio, the volatility estimated by the GARCH(1, 1) model, and the realized volatility.

**Table 1 entropy-26-00192-t001:** CE and correlation coefficient.

**Panel A: Pearson Correlation Coefficient**				**Panel B: Spearman Correlation Coefficient**			
	600,489	601,899	603,993		600,547	601,899	603,993
600,352	0.40	0.44	0.40	600,352	0.31	0.45	0.40
600,426	0.33	0.44	0.46	600,426	0.20	0.49	0.43
600,989	0.34	0.38	0.40	600,989	0.15	0.34	0.39
601,216	0.36	0.43	0.41	601,216	0.17	0.45	0.41
**Panel C: CE**				**Panel D: Industries (from Shenwan)**			
	600,547	601,899	603,993	Metallics:			
600,352	0.04	−0.01	−0.06	600,547, 601,899, 603,993			
600,426	−0.05	−0.07	0.00	Basic chemical industry:			
600,989	−0.09	−0.03	−0.01	600,352, 600,426, 600,989, 601,216			
601,216	−0.01	0.00	0.02				

**Table 2 entropy-26-00192-t002:** Correlation between seven risk metrics (Pearson).

	OR	MR	HR	FR	ORP	EVOL	VOL
OR	1						
MR	0.8840	1					
HR	0.7445	0.9613	1				
FR	−0.9354	−0.9792	−0.8936	1			
ORP	0.5249	0.5285	0.4579	−0.5342	1		
EVOL	0.5861	0.6187	0.5477	−0.6376	0.7975	1	
VOL	0.6215	0.6596	0.5939	−0.6720	0.8225	0.9089	1

All *p*-values are less than 0.0001.

**Table 3 entropy-26-00192-t003:** Variables.

Variable	Explanation	Computation
R	The excess return of a portfolio.	The 20-day return of a portfolio minus 20-day risk-free rate.
C	The curvature.	Ollivier–Ricci, Menger–Ricci, Haantjes–Ricci, and Forman–Ricci.
RF	Risk-free rate.	3-month time deposit rate in China (in the 20-day term).
RM	Market factor.	For two-factor model: the excess return of CSI 300 index relative to the risk-free rate. For four-factor model: download from CSMAR directly. The excess return of the market considering reinvested cash dividends relative to risk-free rate.
SMB	Market value factor.	Download from CSMAR directly. The difference in return between the small-cap portfolio and the large-cap portfolio in A-share market.
HML	Book-to-market factor.	Download from CSMAR directly. The difference in return between the high book-to-market portfolio and the low book-to-market portfolio in A-share market.

**Table 4 entropy-26-00192-t004:** Two-factor model result.

	CE				Pearson Coefficient			
	(1)	(2)	(3)	(4)	(5)	(6)	(7)	(8)
	R	R	R	R	R	R	R	R
BO	−0.03 *				−0.02 *			
	(−1.95)				(−1.83)			
BM		−4.34 ***				−3.97 *		
		(−2.65)				(−1.94)		
BH			−1358.35 ***				−1121.69 **	
			(−2.89)				(−2.00)	
BF				20.19 **				17.78 *
				(2.25)				(1.95)
BRM	0.00	0.00	0.00	0.00	0.01	0.01	0.01	0.00
	(0.24)	(0.34)	(0.45)	(0.31)	(0.74)	(0.74)	(0.86)	(0.68)
_cons	−0.00	−0.00	−0.01	−0.00	−0.01	−0.01	−0.01	−0.01
	(−0.46)	(−0.52)	(−0.79)	(−0.44)	(−0.82)	(−0.82)	(−1.05)	(−0.71)
*N*	29,546	29,546	29,546	29,546	28,045	28,045	28,045	28,045
adj. *R*^2^	4.43%	4.33%	4.21%	4.61%	6.41%	6.22%	5.64%	6.58%

*, **, and *** represent the significance level of 0.1, 0.05, and 0.01, respectively. The values of the associated *t*-test are listed in parentheses. BO, BM, BH, BF, and BRM represent the factor exposure of stocks on the curvature of Ollivier–Ricci, Menger–Ricci, Haantjes–Ricci, Forman–Ricci, and market return, respectively.

**Table 5 entropy-26-00192-t005:** Four-factor model result.

	CE				Pearson Coefficient			
	(1)	(2)	(3)	(4)	(5)	(6)	(7)	(8)
	R	R	R	R	R	R	R	R
BO	−0.02				−0.02			
	(−1.46)				(−1.41)			
BM		−5.05 ***				−3.46		
		(−2.94)				(−1.59)		
BH			−1661.89 ***				−1314.53 **	
			(−3.28)				(−2.10)	
BF				20.10 **				14.51
				(2.26)				(1.59)
BRM	−0.00	−0.00	−0.00	−0.00	0.00	0.00	0.01	0.00
	(−0.27)	(−0.20)	(−0.12)	(−0.20)	(0.70)	(0.78)	(0.83)	(0.72)
BSMB	−0.00	−0.00	−0.00	−0.00	−0.00	−0.00	−0.00	−0.00
	(−0.97)	(−1.06)	(−1.07)	(−1.06)	(−0.65)	(−0.77)	(−0.81)	(−0.74)
BHML	−0.00	−0.00	−0.00	−0.00	−0.00	−0.00	−0.00	−0.00
	(−0.40)	(−0.35)	(−0.29)	(−0.38)	(−0.14)	(−0.22)	(−0.24)	(−0.14)
_cons	−0.00	−0.00	−0.00	−0.00	−0.00	−0.00	−0.00	−0.00
	(−0.01)	(−0.19)	(−0.27)	(−0.05)	(−0.53)	(−0.66)	(−0.77)	(−0.56)
N	19,039	19,039	19,039	19,039	19,276	19,276	19,276	19,276
adj. R^2^	16.57%	16.76%	16.84%	16.71%	14.47%	14.46%	14.02%	14.63%

** and *** represent the significance level of 0.05 and 0.01, respectively. The values of the associated *t*-test are listed in parentheses. BO, BM, BH, BF, BRM, BSMB, and BHML represent the factor exposure of stocks on Ollivier–Ricci curvature, Menger–Ricci curvature, Haantjes–Ricci curvature, Forman–Ricci curvature, market factor, SMB, and HML, respectively.

**Table 6 entropy-26-00192-t006:** Conditional sorting and grouping.

	CE				Pearson Coefficient			
	(1)	(2)	(3)	(4)	(5)	(6)	(7)	(8)
	R	R	R	R	R	R	R	R
1	−0.03 *	−4.67 ***	−1446.44 ***	21.73 **	−0.02	−3.10	−859.74 *	13.74
	(−1.95)	(−2.85)	(−3.18)	(2.39)	(−1.66)	(−1.64)	(−1.69)	(1.62)
2	−0.02	−3.81 **	−1167.05 **	18.15 *	−0.01	−2.08	−455.16	10.45
	(−1.57)	(−2.26)	(−2.53)	(1.98)	(−1.21)	(−1.16)	(−0.94)	(1.31)
3	−0.02	−3.38 **	−1123.39 **	13.98 *	−0.02	−2.87 *	−834.31 *	12.48 *
	(−1.18)	(−2.24)	(−2.52)	(1.73)	(−1.58)	(−1.78)	(−1.86)	(1.69)
4	−0.03 **	−5.06 ***	−1691.93 ***	20.82 **	−0.02 *	−3.44 *	−924.49 *	14.68 *
	(−2.02)	(−3.19)	(−3.60)	(2.49)	(−1.87)	(−1.90)	(−1.81)	(1.84)

*, **, and *** represent the significance level of 0.1, 0.05, and 0.01, respectively. The values of the associated *t*-test are listed in parentheses. Columns 1 and 5 use Ollivier–Ricci curvature, columns 2 and 6 use Menger–Ricci curvature, columns 3 and 7 use Haantjes–Ricci curvature, and columns 2 and 6 use Forman–Ricci curvature.

## Data Availability

All data are from the CSMAR database.

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
