# Peer review of "Vulnerability Analysis Method Based on Network and Copula Entropy"

_entropy, 2024, doi:10.3390/e26030192_

Round 1
Reviewer 1 Report (Previous Reviewer 4)
Comments and Suggestions for Authors
The paper covers a very interesting topic. Its structure is the usual one, each part having enough information, a good quality, and the apropriate lenght and deepness. The list of references covers from the most seminal ones bigining in the first years of the 20th to the most up-to-date. The quality of the english language is good; nevertheless I usually I recommend one more revision. There is no doubt that the added text increased the quality of the paper.
Comments on the Quality of English LanguageThe paper covers a very interesting topic. Its structure is the usual one, each part having enough information, a good quality, and the apropriate lenght and deepness. The list of references covers from the most seminal ones bigining in the first years of the 20th to the most up-to-date. The quality of the english language is good; nevertheless I usually I recommend one more revision. There is no doubt that the added text increased the quality of the paper.
Author Response
Thank you for taking the time to review our article and for your positive feedback and encouragement.
Reviewer 2 Report (Previous Reviewer 3)
Comments and Suggestions for Authors
you claim that you "have rearranged these stocks..."; but in what order ? what are the different types of companies/industries?
x-axis labels on figures; sorry, but your reply/answer is wholly unsatisfactory; I still do not understand what the labels mean; for example "06/2020"; it indicates a month in a year; is the data point resulting from an average for data in the month? a value at the end of a month? at the beginning of a month? Same for Fig. 4 , of course; what is the mention of year meaning?
I fully disagree about the comment justifying your addition of references to previous work; an elementary search ["copula entropy" "financial network"] shows other papers like
Cerqueti, R., Rotundo, G., & Ausloos, M. (2018). Investigating the configurations in cross-shareholding: A joint copula-entropy approach. Entropy, 20(2), 134. Cerqueti, R., & Rotundo, G. (2023). The weighted cross-shareholding complex network: a copula approach to concentration and control in financial markets. Journal of Economic Interaction and Coordination, 18(2), 213-232. Chabot, M., & Bertrand, J. L. (2021). Complexity, interconnectedness and stability: New perspectives applied to the European banking system. Journal of Business Research, 129, 784-800. and I do not list other papers on network resilience in banking and other finance topics Due to these failures, I regret to recommend a rejection of the paper Comments on the Quality of English Languageno comment; grammar and vocabulary can be improved, but the text is quite readable
Author Response
Please see the attachment.

Reviewer 3 Report (Previous Reviewer 1)
Comments and Suggestions for Authors
My earlier comments are fully addressed. This is now ready for publication.
Comments on the Quality of English LanguageFine
Author Response
Thank you for taking the time to review our article and for your positive feedback and encouragement.
This manuscript is a resubmission of an earlier submission. The following is a list of the peer review reports and author responses from that submission.
Round 1
Reviewer 1 Report
Comments and Suggestions for Authors
This paper presents a new means of measuring risk based on Ricci curvature, itself calculated using Copula Entropy (CE) as a distance measure. The paper is clearly written (with some exceptions as noted below) but I was unconvinced that the new formalism it introduced was a major improvement over existing ways of measuring market risk.
This paper is framed in terms of financial instability but calculations are about using differential geometry methods as a better way to compute linkages between market assets within and across .
There are certainly networks between financial institutions – of loans – so that defaults can cascade across the network. But the application was to understanding correlation in stock markets. To be certain there is a linkage – the idea of correlation breakdown during periods of market turmoil is certainly of relevance. Diversifying across stocks within a sector and across various sectors (and markets) is a, or perhaps even the key, risk management tool for portfolio investors. using a more advanced method than standard linear correlation to investigate the way in which the network of connections between sectors would be of interest.
This is how I would have done the motivation and economic framing of the paper. Then the results of the calculations would fit the motivation. Right now it does not. I think this is easily enough changed and would greatly improve the paper and so:
Recommendation 1: The reviewers should frame the paper in terms of financial markets, not in terms of the banking sector.
Another thing that I found, I think needlessly, confusion was the discussion of copula entropy (CE) and description of how that fits with mutual information. All true but what did it have to do with the rest of the paper? The initial description (Section 2.2) of Ricci curvature seemed disjointed from the description (Section 2.1) of CE, although later it emerged that the CE was playing the role of the distance metric in the calculations. This in itself is confusing to me, as I had thought that mutual information was in fact not a valid metric function because it was not symmetric in its arguments.
That point should be addressed.
Recommendation 2: The authors should discuss whether CE is a distance metric and, if not, why it can still be used in a Ricci Curvature calculation as they do.
Then the results of the paper are summarizing.
Comparing correlation across sectors with Pearson and Spearman Correlations (Figure 1): Recommendation 3: Certainly the results depicted are different. How do you know they are “better”? Better in what sense? Either you need more discussion here, or if the point is simply that your new method is measuring different things than the Spearman or Pearson coefficients, then this is clearly showing that and just claim that.
Four ways of discretizing the curvature metric (Figure 2):
Recommendation 4:
Suggest plotting -1* the Forman curvature to make more all the same shape.
Here the difference with a traditional correlation metric driven curvature is not so marked. Again I’d like to see a bit more discussion of why this is better
The way the clustering works (Figure 3)
This is really interesting. On what basis were the four particular dates at which the network structure was displayed I am struck by how different the networks seem to be on the two dates 2014-01-03 and 2017-02-17 vs the other times. Is this a difference in reality or a difference in display?
Recommendation 5: Discuss the above points.
Comparing with volatility driven risk metrics (Figure 5)
Recommendation 6: Again, suggest plotting -1* the Forman curvature to make all roughly comparable.
Recommendation 7: Both in the visual inspection of Figure 5 and the analysis of Table 1 confirm: All seven of the risk measures are broadly aligned. That is good and not so surprising. But what is the value added of the new approach then? The statement which ends the paper, which suggests that it’s an OK risk metric to use too, is fine. But unsatisfying. The other volatility-based metrics are time tested and practitioners have a sense of what they mean. The new approach has to bring something new to the table.
References
Recommendation 8: Reference [12] seems a strange reference for the statement it supports, that copula entropy is the same as interaction entropy except with the opposite sign. Is there a better way to support this result?
Comments on the Quality of English LanguageMinor and typographical
Overall the paper is well written and the English is good. I haven’t read it carefully for minor points but
Line 235 “bearing the bear market”: Suggest rewording to “enduring the bear market”
view
Reviewer 2 Report
Comments and Suggestions for Authors
This is a very detailed study of risk indicators for market instability.
Its main weaknesses are, in my opinion:
1) It follows rather one-to-one, and down into the very details, the study by
A. Samal et al. in Royal Society Open Science, 2021 (their Ref. [15]). In particular, exactly the same four versions of Ricci curvature for graphs are considered; the relationship between distances and edge weights (Eq.(10)) is the same; the same conventional risk measures including the GARCH model are used for comparison; the data are presented and evaluated in exactly the same way.
Nevertheless, these analogies with Ref. [15] are not acknowledged, which I consider as serious plagiarism.
2) The main differences with Ref. [15] are:
--Instead of the S\&P 500 and Nikkei-225 indices [15], the authors used the
CSI 300 index, without however saying what this is (the biggest Chinese index), nor where it can be found.\\
-- Instead of using the Pearson correlation for a similarity measure of network nodes [15], they use the mutual information.
The second aspect (using MI) is, as far as I can see, the main novelty of the paper. It would be very interesting, provided the authors would have made a careful comparison with other (Pearson or Spearman correlations). Referring to their Figure 1, they claim ``As can be seen from the figure, CE provides a clearer correlation pattern than these two correlation coefficients}".
Unfortunately, I cannot see anything like this.
On the other hand, a comparison could have been made with the results of [15]. But since the authors chose to use different stock indices, such a comparison is impossible and was not even attempted.
3) Throughout the paper, the author call the MI ``{\it copula entropy}", following Ref.[9]. But it was known long before [9] that MI can be estimated using copulas, see e.g. their Ref. [14]. Even though the word `copula' is not used in [14], the use of ranks instead of original data advocated in that paper corresponds exactly to the use of copulas. `Copulas' were not mentioned, because these (and previous) authors considered Sklar's theorem as an obvious triviality (the replacement of data by ranks is obviously monotonic, and MI is invariant under such monotonic transformations of marginals).
4) Table 1 is a bit strange. On the one hand, what are the entries ``(0.0000)"?
On the other hand, why defining ``*" and `**", if they are not used, since all entries have p-values < 0.01?
5) The English would need some improvements.
Some formulations are clumsy, and some articles are wrong.
Reviewer 3 Report
Comments and Suggestions for Authors
this paper seems to be already freely available on the web
i do not understand the columns and rows of fig.1
strange way of inserting references in the text
no consistence in fact; awful
funny to read Mark NewMan; fortunately it is not yet Mark New Man; and he is a male, good for that
i do not understand the label on the x-axis in fig.2, fig. 3, fig.5; are these days of the month? which one ?
fig.4 : since there are no labels nor explanation in the caption, I 'm wondering about the educational quality of the authors
the statistical validity of the results (table 1) is not convincing, whence the conclusion
bibliography. literature review : there are many other ( even recent papers, even published in Entropy). about "copula entropy; networks; ", as one can find at once on Google Scholar
Comments on the Quality of English Languagestrange way of writing references
Reviewer 4 Report
Comments and Suggestions for Authors
The paper "Vulnerability analysis method based on network and Copula entropy" covers a very interesting topic. The structure follows the usual patern with very section being clearly written, understandablehaving and equilibrated in terms of content.
In my opinion it has all the quality requisites to be published after a very small revision.

Round 2
Reviewer 1 Report
Comments and Suggestions for Authors
Good to go!
Comments on the Quality of English LanguageGood to go
Reviewer 2 Report
Comments and Suggestions for Authors
The authors have responded marginally to the points raised in my first report.
1) Mutual information is still called "copula entropy". Instead of the detailed discussion of copulas, the authors could just have said that they evaluated MI by using ranks of the marginals, giving a reference e.g. to [24] or any other of the many references prior to [16,21].
2) When referring to Ref. [25] (Samal et al.), no hint is given that the present work is basically a reproduction of it.
3) Table 1 could be greatly simplified by just saying in the head line that all p-values are < 0.0001.
4) It is still not said what "CSI" is, and where these data can be found.
5) The comparison between MI, Pearson, and Spearman coefficients still gives no clear hint which of them is superior.
6) There is still no comparison of the results with those of [25], which would have been essential for appreciating that MI is superior. To compare with [25], the authors shouls have applied their analyses also to the S&P or Nikkei indices, which is still not done.
I still cannot recommend publication.
Comments on the Quality of English Language
Some formulations are improved.
Round 3
Reviewer 2 Report
Comments and Suggestions for Authors
Again, the changes made by the authors are insufficient.
1) The authors still use "copula entropy" in the title, the abstract, and throughout the paper, instead of saying that the use mutual entropy, and that they used ranks instead of original data in the marginals. This was wide spread usage before the papers by Ma & Sun [16,21]. Of course they can can cite Ma & Sun [16,21] and say that their algorithm is equivalent to using copulas, but even then the paper could be shortened considerably, and readers would would know from the beginning what is the aim of the paper.
In lines 132-134, the authors say "From this, it can be understood that Copula entropy and mutual information share a fundamental unity as they both quantify the amount of information regarding all orders of correlation between random variables." Sorry, but this can be said more simply: Copula entropy is mutual information.
2) The authors now say " We adopt a similar analysis route with
Samal et al. [26]. But this practice is relatively general...". No, apart from replacing correlations by mutual information and applying it to a different stock index, they use **exactly** the same method as Samal et al. [26]. And that paper was highly innovative and not standard at all.
3) I cannot see from Fig.2 that MI ("CE") is systematically better that Pearson, as claimed by the authors. For Haantjes curvature, Pearson seems to show more structure than MI. But a more detailed analysis is not possible, because of the insufficient figure caption. For each curvature, there are two plots. Superficially they look identical (except for Forman, where the signs are inverted), but I guess they are not -- why would they be then shown twice?
4) The latter problem occurs also for Fig.5: Why are all plots shown twice? What is anyhow the difference between Figs. 2 and 5? (apart from the fact that in Fig.2 also Pearson-based results are shown)
5) The same doubling occurs also in Fig.1. The authors claim that the visible difference between the MI-based and the correlation-based plots show that the former are better. But this difference results mainly from the fact that MI is basically analogous to the **square** of correlation coefficients. Thus light regions in the correlation-based plots, where correlations are small, should be even lighter in the MI-based plot. This doesn't mean that either of them is better. Table 1 on page 7 is thus irrelevant.
5) Why are there still all these stars in the other Table 1 on page 15?
6) If I understand correctly, "curvatures" of networks are strongly correlated with average connectedness (i.e., average node degrees). High average degree implies high curvature, as also seen from Fig.3. Do the authors just find that the economy is vulnerable when returns of different assets are in average strongly correlated? That would make sense intuitively, and I wonder whether it is widely accepted.
7) For estimating correlation coefficients or MI, one needs for each of the stochastic variables a number of "events" (typically > 10 for Pearson correlations, and even more for any non-trivial MI estimate). The authors show time-resolved results, i.e. they cannot use for this the entire time sequences, but they have to use windows centered at the nominal times for which they quote the MI. How large windows were actually used?
Comments on the Quality of English Language
None
